# Hierarchical Lattice Layer for Partially Monotone Neural Networks

**Hiroki Yanagisawa**
IBM Research - Tokyo
IBM Japan, Ltd.
Tokyo, Japan
yanagis@jp.ibm.com

**Kohei Miyaguchi**
IBM Research - Tokyo
IBM Japan, Ltd.
Tokyo, Japan
miyaguchi@ibm.com

**Takayuki Katsuki**
IBM Research - Tokyo
IBM Japan, Ltd.
Tokyo, Japan
kats@jp.ibm.com

## Abstract

Partially monotone regression is a regression analysis in which the target values are monotonically increasing with respect to a subset of input features. The TensorFlow Lattice library is one of the standard machine learning libraries for partially monotone regression. It consists of several neural network layers, and its core component is the lattice layer. One of the problems of the lattice layer is that it requires the projected gradient descent algorithm with many constraints to train it. Another problem is that it cannot receive a high-dimensional input vector due to the memory consumption. We propose a novel neural network layer, the hierarchical lattice layer (HLL), as an extension of the lattice layer so that we can use a standard stochastic gradient descent algorithm to train HLL while satisfying monotonicity constraints and so that it can receive a high-dimensional input vector. Our experiments demonstrate that HLL did not sacrifice its prediction performance on real datasets compared with the lattice layer.

## 1   Introduction

In regression analysis, we often have prior knowledge that the target values are monotonically increasing (or decreasing) with respect to a subset of input features. For example, when we estimate house prices from house data (e.g., the area size, the number of rooms, and the distance to the nearest supermarket), we know that the target value (i.e., the house price) is monotonically increasing with respect to some of the input features (e.g., the area size) if the other features are equal [Potharst and Feelders, 2002].

*Partially monotone regression* is a regression analysis that exploits such prior knowledge. The input of this regression is a feature vector $x \in \mathbb{R}^d$, which can be partitioned into $x = (p, q)$ such that $p \in \mathbb{R}^{d-m}$ is non-monotone features and $q \in \mathbb{R}^m$ is monotone features. The goal of this regression is to find a function $f$ such that $f(x) = f(p, q) \in \mathbb{R}$ is monotonically increasing on $q$. Partially monotone regression models have advantages over the standard regression models. For example, they have a better regularization capability [Dugas *et al.*, 2000; Fard *et al.*, 2016; You *et al.*, 2017] and better interpretability [Gupta *et al.*, 2016]. They can be used for fair machine learning [Wang and Gupta, 2020].

Many machine learning models have been developed for partially monotone regression. For example, there are decision-tree based models [Potharst and Feelders, 2002], SVM models [Lauer and Bloch, 2008], boosting models [Bartley *et al.*, 2018], and random forest models [Bartley *et al.*, 2019]. Many popular open source libraries for gradient boosting can handle partial monotonicity (e.g., CatBoost, LightGBM, and XGBoost). See a survey paper [Cano *et al.*, 2019] and the references in [Gupta *et al.*, 2016] for comprehensive surveys.

36th Conference on Neural Information Processing Systems (NeurIPS 2022).

As for neural network models, the TensorFlow Lattice library [Canini *et al.*, 2016; Gupta *et al.*, 2016; You *et al.*, 2017] is a standard model for partially monotone regression, and it is available as an open source library. It consists of several neural network layers, and its core component is the lattice layer [Gupta *et al.*, 2016]. Hereafter, we call this layer TL Lattice. TL Lattice constructs a hypercube with $k^d$ vertices as an internal structure, where $k \geq 2$ is an integer hyperparameter for specifying the granularity of the hypercube, and $d$ is the dimension of the input vector $x \in \mathbb{R}^d$. One of its problems is its memory consumption in storing $k^d$ parameters, and therefore, it cannot receive a high-dimensional input vector with a large $d$. This means that we need to combine multiple neural network layers to handle an input vector with a large $d$ (e.g., by using small lattices [Canini *et al.*, 2016; You *et al.*, 2017]). Another problem is its requirement for the projected gradient descent algorithm with many constraints to train it to handle monotonicity constraints. This means that we cannot use a standard stochastic gradient descent algorithm to train it. The details of the training algorithms used for the TensorFlow Lattice library are described in [Gupta *et al.*, 2016; You *et al.*, 2017].

In this paper, we propose a novel neural network layer, the *hierarchical lattice layer* (HLL), as an extension of TL Lattice. The internal structure of HLL is similar to that of TL Lattice in that both layers construct hypercube structures. However, they are different in several aspects. For example, HLL constructs an $m$-dimensional hypercube, whereas TL Lattice constructs a $d$-dimensional hypercube, where $d$ and $m$ are the dimensions of the input vector $x \in \mathbb{R}^d$ and its monotone feature $q \in \mathbb{R}^m$, respectively. Since $m$ is usually a small integer in practice, the number of vertices $k^m$ in HLL is significantly smaller than that of TL Lattice (i.e., $k^d$). Therefore, HLL can receive a high-dimensional input vector as long as $m$ is small. For another example, the parameterization of HLL is designed to represent a class of partially monotone functions (i.e., HLL cannot represent a function $f(p, q)$ that is not monotonic on $q$) by using the hierarchical relationship between the vertices of the hypercube. Therefore, the internal structure of HLL ensures partial monotonicity regardless of the training algorithm used. In contrast, the internal structure of TL Lattice is a lookup table, and the table itself can represent any function $f(p, q)$, which means that the parameterization of TL Lattice is not designed to satisfy monotonicity constraints.

In our experiments, we demonstrate that HLL did not sacrifice its prediction performance compared with TL Lattice. Note that HLL is designed to satisfy monotonicity constraints and is not designed to improve the prediction performance compared with TL Lattice. Therefore, the purpose of our experiments was not to show improvements in prediction performance compared with TL Lattice but to show that HLL did not sacrifice the prediction performance to satisfy the constraints. In addition, we demonstrate that we could use HLL for high-dimensional input vectors with a small $m$.

## 2 Partially Monotone Regression

Regression analysis is a method for finding a function $f : \mathbb{R}^d \to \mathbb{R}$ such that $f(x_i)$ approximates $y_i$ for a given dataset $D = \{(x_i, y_i)\}_{i=1}^n$, where $x_i \in \mathbb{R}^d$ is a feature vector, and $y_i \in \mathbb{R}$ is a target value. In partially monotone regression, we assume that we can partition a feature vector $x \in \mathbb{R}^d$ into $x = (p, q) \in \mathbb{R}^{d-m} \times \mathbb{R}^m$, where $0 < m < d$, such that $f(x) \in \mathbb{R}$ is monotonic on $q$. A function $f(x)$ for $x = (p, q)$ is *monotonic* on $q$ if this inequality holds:

$$f(p, q) \leq f(p, q'), \ \forall p, \ \forall q \leq q',$$

where $q \leq q'$ denotes the inequality for all the elements (i.e., $q[i] \leq q'[i]$ for all $1 \leq i \leq m$, where $q[i]$ denotes the $i$-th element of $q$).

Partially monotone regression has many applications in various fields, and it has important applications even if $m = 1$. An example of such application with $m = 1$ is quantile regression [Koenker and Bassett, 1978; Zhou *et al.*, 2020], which is used to estimate the conditional quantiles of a target variable distribution. We show that the output $q(x) = q(x', \tau)$ of the quantile regression for an input $x = (x', \tau)$ is monotonic on $\tau$, where $x' \in \mathbb{R}^{d-1}$ is a feature vector, and $\tau \in [0, 1]$ is a quantile level. In quantile regression, we estimate the quantile value $q(x', \tau)$ defined by

$$q(x', \tau) = \inf\{y \mid F(y|X = x') \geq \tau\},$$

where $F(y|X = x')$ is the conditional cumulative distribution function of $Y$ given $X = x'$:

$$F(y|X = x') = \Pr(Y \leq y|X = x').$$

Since the function $q(x', \tau)$ is monotonically increasing on $\tau$, this problem can be modeled as partially monotone regression with $m = 1$. Note that, if we use a standard multi-layer perceptron without any monotonicity constraints to estimate the quantile value $q(x', \tau)$, it sometimes fails to satisfy the monotonicity constraints, which is known as the *crossing problem* (see e.g., [Tagasovska and Lopez-Paz, 2019]). In quantile regression, many algorithms have been proposed to resolve the crossing problem (e.g., [Park *et al.*, 2022]).

Another example of partially monotone regression with $m = 1$ is survival analysis, which is also known as time-to-event analysis [Wang *et al.*, 2019; Rindt *et al.*, 2022]. We show that the output $f(x) = f(x', t)$ of the survival analysis for an input $x = (x', t)$ is monotonic on $t$, where $x' \in \mathbb{R}^{d-1}$ is a feature vector of a patient, and $t \in \mathbb{R}$ is the time of an event of interest of this patient. In survival analysis, the event of interest typically corresponds to the death of a patient. The goal of the analysis is to estimate a survival function $S(x', t)$ for patient $x'$, which represents the probability that the event does not occur until time $t$ for patient $x'$. By setting $f(x', t) = 1 - S(x', t)$, we can model the survival analysis as a partially monotone regression with $m = 1$, where the value $f(x', t)$ shows the mortality rate of patient $x'$ at time $t$, and $f(x', t)$ is monotonically increasing on $t$.

## 3 Hierarchical Lattice Layer

In this section, we explain our new *hierarchical lattice layer* (HLL) for neural network models for partially monotone regression. It can be seen as an extension of TL Lattice [Gupta *et al.*, 2016]. Note that the input of HLL is a $d$-dimensional vector $x \in [0, 1]^d$, and its output $f(x)$ is in $[0, 1]$, where the input vector $x$ can be partitioned into two subvectors $p \in [0, 1]^{d-m}$ and $q \in [0, 1]^m$ such that $f(x)$ is monotonic on $q$.

We create the $m$-dimensional unit hypercube $[0, 1]^m$ as an internal structure of HLL, and we decompose it into $(k_1 - 1) \times (k_2 - 1) \times \cdots \times (k_m - 1)$ subhypercubes by dividing the interval $[0, 1]$ of the $i$-th dimension into $k_i - 1$ subintervals, where each $k_i \geq 2$ is an integer hyperparameter that specifies the granularity of the hypercube. Unless otherwise stated, we assume that $k_1 = k_2 = \cdots = k_m = k$, and the interval $[0, 1]$ is divided into $k - 1$ equal-length subintervals just for simplicity. This means that the length of an edge of a subhypercube is $1/(k - 1)$, and we have the $k^m$ vertices in the unit hypercube. We denote the set of the vertices in the unit hypercube by $V_m$. In this paper, we do not distinguish between a vertex $v \in V_m$ and its coordinate $q_v \in [0, 1]^m$. This means that the value $f(p, q_v)$ for coordinate $q_v$ is written as $f(p, v)$.

### 3.1 Key Idea of HLL

Before describing the details of HLL, we explain the key idea of HLL by using a simple case with $m = 1$ and $k = 3$. In this case, HLL is supposed to learn a function $f(p, q)$ such that $0 \leq f(p, q_0) \leq f(p, q_1) \leq f(p, q_2) \leq 1$ holds for any $p \in \mathbb{R}^{d-m}$ and $q_1 = 0$, $q_2 = 1/2$, and $q_3 = 1$. Hereafter, we use $v_1$, $v_2$, and $v_3$ to represent $q_1$, $q_2$, and $q_3$, respectively, to emphasize that they correspond to the vertices in $V_m$.

In HLL, we associate a function $g(p, v)$ for each vertex $v \in V_m$ so that the function $f(p, v)$ for $v \in V_m$ can be represented by using the function $g(p, v)$. We show how the function $g(p, v)$ is defined here. First, the function $g(p, v_0)$ is defined as $g(p, v_0) = f(p, v_0)$. Second, the function $g(p, v_1)$ is defined as the function that satisfies this equation:
$$f(p, v_1) = (1 - g(p, v_1))f(p, v_0) + g(p, v_1).$$
Note that $0 \leq g(p, v_1) \leq 1$ always holds because $f(p, v_0) \leq f(p, v_1) \leq 1$. Similarly, the function $g(p, v_2)$ is defined as the function that satisfies this equation:
$$f(p, v_2) = (1 - g(p, v_2))f(p, v_1) + g(p, v_2).$$
See Figure 1 for an illustration of these definitions. By using these definitions, the task of learning the function $f(p, v)$ with monotonicity constraints $0 \leq f(p, v_0) \leq f(p, v_1) \leq f(p, v_2) \leq 1$ is casted as the task of learning the function $g(p, v)$ such that $0 \leq g(p, v) \leq 1$ without any monotonicity constraints. We generalize this idea for $m > 1$ in the following.

We note here that HLL for the case with $m = 1$ is similar to some existing approaches used in survival analysis (see e.g., [Ren *et al.*, 2019; Zheng *et al.*, 2019]). More specifically, the survival function is decomposed into a product of hazard rates in these approaches, and the survival function and the hazard rate can be represented by using $f(p, v)$ and $g(p, v)$ in HLL, respectively.

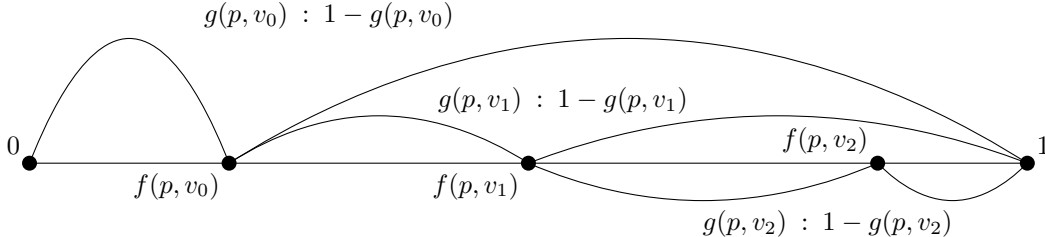

Figure 1: Illustration of parameterization of HLL for $m = 1$ and $k = 3$.

## 3.2 Hierarchical Relationship between Vertices

In HLL, we assume that there is a total order $\sigma = (v_0, v_1, \ldots, v_{|V_m|-1})$ of vertices in $V_m$. We can use any total order $\sigma$ to construct HLL, but unless otherwise stated, we assume that $\sigma$ is determined by the increasing order of the Manhattan distance (i.e., $L_1$ distance) from the origin vertex $v_0 = (0, 0, \ldots, 0) \in [0, 1]^m$ of the unit hypercube with an arbitrary tie-breaker. See Figure 2 for an example of a decomposed unit hypercube with $m = 2$ and $k = 4$ and the total order $\sigma = (v_0, v_1, \ldots, v_{15})$ determined by the Manhattan distance.

We define the *domination* relationship between vertices in $V_m$, and we define the minimal dominated and dominating sets based on $\sigma$.

**Definition 3.1** *(Domination.) A vertex $v \in \mathbb{R}^m$ is said to dominate another vertex $u \in \mathbb{R}^m$ if and only if $u[i] \leq v[i]$ holds for all $i \in \{1, 2, \ldots, m\}$ and there exists an index $i' \in \{1, 2, \ldots, m\}$ such that $u[i'] < v[i']$.*

**Definition 3.2** *(Minimal dominated set.) The set of vertices $L_{v,\sigma}$ denotes the minimal set of vertices that satisfy three conditions (i) every vertex $u \in L_{v,\sigma}$ is dominated by $v$, (ii) vertex $u$ precedes $v$ in $\sigma$, and (iii) no other vertex $w \in L_{v,\sigma}$ dominates $u$.*

**Definition 3.3** *(Minimal dominating set.) The set of vertices $U_{v,\sigma}$ denotes the minimal set of vertices that satisfy three conditions (i) every vertex $u \in U_{v,\sigma}$ dominates $v$, (ii) vertex $u$ precedes $v$ in $\sigma$, and (iii) no other vertex $w \in U_{v,\sigma}$ is dominated by $u$.*

Intuitively the domination relation means that $f(p, u) \leq f(p, v)$ holds for any $p \in \mathbb{R}^{d-m}$ if vertex $v$ dominates vertex $u$. Regarding the minimal dominated and dominating sets, there can be multiple minimal sets that satisfy the conditions. Throughout this paper, we fix an (arbitrary) minimal dominated set $L_{v,\sigma}$ and an (arbitrary) minimal dominating set $U_{v,\sigma}$ for each $v \in \sigma$. Note also that we have $|L_{v,\sigma}| \leq m$ and $|U_{v,\sigma}| = 0$ if $\sigma$ is determined by the Manhattan distance. This is why we use the Manhattan distance to determine $\sigma$, because we have weaker upper bounds $|L_{v,\sigma}| \leq |V_m|$ and $|U_{v,\sigma}| \leq |V_m|$ if $\sigma$ is an arbitrary ordering.

**Example.** In the hypercube shown in Figure 2, vertex $v_8$ dominates the vertices in $\{v_0, v_1, v_2, v_4, v_5\}$ and is dominated by the vertices in $\{v_{11}, v_{12}, v_{13}, v_{14}, v_{15}\}$. By the definition of the minimal dominated set, we have $L_{v_8,\sigma} = \{v_4, v_5\}$ because vertices $v_0, v_1,$ and $v_2$ are dominated by $v_4$ or $v_5$. By the definition of the minimal dominating set, we have $U_{v_8,\sigma} = \emptyset$ because all the vertices that dominate $v_8$ appear after $v_8$ in ordering $\sigma$.

## 3.3 Auxiliary Neural Network

We explain the function $g(p, v)$ associated with each vertex $v \in V_m$. To define it, we define two functions $l(p, v)$ and $u(p, v)$ for $p \in \mathbb{R}^{d-m}$ and a vertex $v \in V_m$ as

$$l(p, v) = \begin{cases} \max\limits_{v' \in L_{v,\sigma}} f(p, v') & \text{if } L_{v,\sigma} \neq \emptyset, \\ 0 & \text{if } L_{v,\sigma} = \emptyset, \end{cases} \tag{1}$$

$$u(p, v) = \begin{cases} \min\limits_{v' \in U_{v,\sigma}} f(p, v') & \text{if } U_{v,\sigma} \neq \emptyset, \\ 1 & \text{if } U_{v,\sigma} = \emptyset, \end{cases} \tag{2}$$

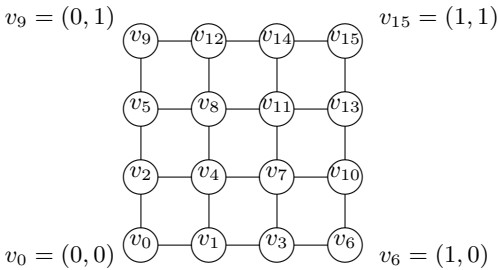

Figure 2: Example of vertices in unit hypercube in HLL with $m = 2$ and $k = 4$.

where $L_{v,\sigma}$ and $U_{v,\sigma}$ are the minimal dominated and dominating sets, respectively. Note that $l(p, v)$ and $u(p, v)$ are defined so as to satisfy the inequality $l(p, v) \leq f(p, v) \leq u(p, v)$. By using this inequality, we define $g(p, v)$ as a function that satisfies

$$f(p, v) = (1 - g(p, v))l(p, v) + g(p, v)u(p, v). \tag{3}$$

Since we have $l(p, v) \leq f(p, v) \leq u(p, v)$, there always exists $0 \leq g(p, v) \leq 1$ that satisfies Eq. (3).

We construct an auxiliary neural network $g_\theta(p, v)$ in HLL to learn the function $g(p, v)$. To satisfy the constraint $0 \leq g(p, v) \leq 1$, we use the sigmoid function in the last layer of $g_\theta(p, v)$ as the activation function. Note that, by the definitions of the three functions $l(p, v)$, $u(p, v)$, and $g(p, v)$, we can compute $f(p, v)$ for any $p$ and $v$ by using $g_\theta(p, v)$.

**Example of Parameterization of HLL.** We show an example of the functions $l(p, v)$, $u(p, v)$, and $g(p, v)$ for a simple case with $m = 1$ and $k = 3$ (see Figure 1). In this case, the hypercube has three vertices $V_m = \{v_0, v_1, v_2\}$, and we use the total ordering $\sigma = (v_0, v_1, v_2)$ here. For the first element $v_0$ of $\sigma$, we have $l(p, v_0) = 0$ and $u(p, v_0) = 1$ because $L_{v_0,\sigma} = U_{v_0,\sigma} = \emptyset$ and $g(p, v_0) = f(p, v_0)$, which satisfies $l(p, v_0) \leq f(p, v_0) \leq u(p, v_0)$ and $0 \leq g(p, v_0) \leq 1$. For the second element $v_1$ of $\sigma$, we have $l(p, v_1) = f(p, v_0)$ and $u(p, v_1) = 1$ because $L_{v_1,\sigma} = \{v_0\}$ and $U_{v_1,\sigma} = \emptyset$. The function $g(p, v_1)$ is defined as a function that satisfies

$$f(p, v_1) = (1 - g(p, v_1)) \cdot l(p, v_1) + g(p, v_1) \cdot u(p, v_1).$$

Note that this definition satisfies $l(p, v_1) \leq f(p, v_1) \leq u(p, v_1)$ and $0 \leq g(p, v_1) \leq 1$, which means that the monotonicity constraint $f(p, v_0) \leq f(p, v_1) \leq 1$ is satisfied. Similarly, for the third element $v_2$ in $\sigma$, we have $l(p, v_2) = f(p, v_1)$ and $u(p, v_2) = 1$ because $L_{v_2,\sigma} = \{v_1\}$ and $U_{v_2,\sigma} = \emptyset$. The function $g(p, v_2)$ is defined as a function that satisfies

$$f(p, v_2) = (1 - g(p, v_2)) \cdot l(p, v_2) + g(p, v_2) \cdot u(p, v_2).$$

Note that this definition satisfies $l(p, v_2) \leq f(p, v_2) \leq u(p, v_2)$ and $0 \leq g(p, v_2) \leq 1$, which means that the monotonicity constraint $f(p, v_1) \leq f(p, v_2) \leq 1$ is satisfied.

### 3.4 Inference and Training Algorithms

The inference algorithm of HLL is similar to TL Lattice, and Figure 3 illustrates it. Given an input $x = (p, q)$, we first calculate $f(p, v)$ for all $v \in V_q$ by using $g_\theta(p, v)$, where $V_q \subseteq V_m$ is the set of the vertices that surround $q$ in the hypercube. Then, we estimate $f(p, q)$ as an interpolation of the values $f(p, v)$ for $v \in V_q$. As for the interpolation algorithm, we can use multilinear interpolation or simplex interpolation [Gupta *et al.*, 2016]. Note that the size of $V_q$ depends on the interpolation algorithm (i.e., $|V_q| = 2^m$ if we use multilinear interpolation, and $|V_q| = O(m \log m)$ if we use simplex interpolation).

The training algorithm of HLL is the same as that of the standard neural network layer. For example, if we train a neural network model that consists of a single HLL, the parameters of HLL (i.e. the parameters $\theta$ of the auxiliary neural network $g_\theta$) are trained to minimize the empirical loss

$$\min_{g_\theta} \sum_{i=1}^{n} \ell(y_i, f(x_i)),$$

Figure 3: Illustration of inference algorithm of HLL for case with $m = 3$, and $q$ is surrounded by vertices $V_q = \{v_1, v_2, \ldots, v_8\}$.

where $\{(x_i, y_i)\}_{i=1}^{n}$ is a dataset, $f(x_i)$ is the output of the inference algorithm of HLL for input $x_i$, and $\ell$ is a loss function.

### 3.5 Partial Monotonicity

As already proven in [Gupta *et al.*, 2016], the function $f(p, q)$ computed by an interpolation algorithm satisfies monotonicity constraints if the values corresponding to the vertices of the hypercube satisfy the constraints. Therefore, we show that we have $f(p, u) \leq f(p, v)$ for any pair of vertices $u, v \in V_m$ such that $u$ is dominated by $v$. Note that this proposition holds for any vertex ordering $\sigma$ and is not restricted to the vertex ordering determined by the Manhattan distance.

**Proposition 3.4** *In HLL, $f(p, u) \leq f(p, v)$ holds for any $p \in \mathbb{R}^{d-m}$ and any pair of vertices $u, v \in V_m$ such that $u$ is dominated by $v$*

*Proof.* Suppose that $u$ precedes $v$ in the total ordering $\sigma$. By the definition of the minimal dominated set, there exists a sequence of vertices $u_0(= u), u_1, u_2, \ldots, u_J(= v)$ such that $u_{j-1} \in L_{u_j, \sigma}$ holds for $j = 1, 2, \ldots, J$. Since $u_{j-1} \in L_{u_j, \sigma}$, we have $f(p, u_{j-1}) \leq l(p, u_j) \leq f(p, u_j)$ by Eq. (1). Hence, we have $f(p, u) = f(p, u_0) \leq f(p, u_1) \leq \cdots \leq f(p, u_J) = f(p, v)$.

Similarly, suppose that $v$ precedes $u$ in the total ordering $\sigma$. By the definition of the minimal dominating set, there exists a sequence of vertices $v_0(= v), v_1, v_2, \ldots, v_J(= u)$ such that $v_{j-1} \in U_{v_j, \sigma}$ holds for $j = 1, 2, \ldots, J$. Since $v_{j-1} \in U_{v_j, \sigma}$, we have $f(p, v_{j-1}) \geq u(p, v_j) \geq f(p, v_j)$ by Eq. (2). Hence, we have $f(p, v) = f(p, v_0) \geq f(p, v_1) \geq \cdots \geq f(p, v_J) = f(p, u)$. □

### 3.6 Complexity Analysis

We show that the time complexity of the inference algorithm of HLL is $O((T(g_\theta) + m)|V_m|)$, where $T(g_\theta)$ is the time complexity of computing $g_\theta(p, v)$ for an input $(p, v) \in \mathbb{R}^{d-m} \times \mathbb{R}^m$. First, the time complexity to compute $f(p, v)$ for all vertices $v \in V_m$ is $O((T(g_\theta) + m)|V_m|)$ since it takes $O(T(g_\theta) + |L_{v, \sigma}| + |U_{v, \sigma}|)$ time to compute $f(p, v)$ for a single vertex $v \in V_m$, and we have $|L_{v, \sigma}| + |U_{v, \sigma}| \leq m$. Then, the time complexity for the interpolation algorithm is $O(2^m)$ if we use multilinear interpolation and $O(m \log m)$ if we use simplex interpolation [Gupta *et al.*, 2016]. Since we have $2^m \leq |V_m|$, the time complexity for the interpolation algorithm is smaller than that for the computation of $f(p, v)$. Hence, the overall time complexity remains $O((T(g_\theta) + m)|V_m|)$. If we can assume that $m$ is a small integer (i.e., $m$ is a constant), the computation time for inference can be written simply as $O(T(g_\theta))$. Note that, if the time complexity $T(g_\theta)$ is a linear function on $d$, the time complexity of the inference algorithm for HLL is smaller than that for TL Lattice because the inference algorithm of TL Lattice takes $O(2^d)$ and $O(d \log d)$ times if we use multilinear interpolation and simplex interpolation, respectively [Gupta *et al.*, 2016].

The space complexity of HLL is $O(|g_\theta| + m|V_m|)$ if the vertex ordering $\sigma$ is determined by the Manhattan distance, where we denote the size of the auxiliary neural network $g_\theta(p, v)$ by $|g_\theta|$, and the $O(m|V_m|)$ term is the space required to store $L_{v, \sigma}$ and $U_{v, \sigma}$ for all $v \in V_m$. Note that, if we use an arbitrary vertex ordering $\sigma$, we require $O(|V_m|^2)$ space to store $L_{v, \sigma}$ and $U_{v, \sigma}$ for all $v \in V_m$ because we know only that $|L_{v, \sigma}| \leq |V_m|$ and $|U_{v, \sigma}| \leq |V_m|$. Since $|V_m|^2$ is much larger than $m|V_m|$, we recommend using the Manhattan distance to determine $\sigma$ in HLL.

# 4 Experiments

In our experiments, we demonstrate that HLL did not sacrifice its prediction performance compared with TL Lattice. Note that HLL is designed to satisfy monotonicity constraints and is not designed to improve the prediction performance compared with TL Lattice. Therefore, the purpose of our experiments was not to show such improvements but to show that HLL did not sacrifice the prediction performance to satisfy the constraints. In addition, we demonstrate that we could use HLL for high-dimensional input vectors with a small $m$. All our experiments were conducted on a virtual machine with an Intel Xeon CPU (3.30 GHz) processor without any GPU and 64 GB of memory running Red Hat Enterprise Linux Server 7.6.

## 4.1 Neural Network Models

In our experiments, we compared these four neural network models.

- **MLP**. A multi-layer perceptron model with three hidden layers. We used the ReLU activation layer for the hidden layers and the sigmoid function for the output layer. Note that this neural network is not designed for partially monotone regression and cannot handle monotonicity constraints.

- **HLL**. A neural network model consisting of a single hierarchical lattice layer. We used an MLP model with three hidden layers for the auxiliary neural network $g_\theta(p, v)$, and we used multilinear interpolation in the inference algorithm.

- **TL Lattice**. A neural network model consisting of a single TL Lattice layer [Gupta *et al.*, 2016]. This layer uses $O(k^d)$ space, and therefore, it can be used only for low-dimensional inputs (i.e., $d$ is a small integer). We used multilinear interpolation in the inference algorithm.

- **TL RTL.** A neural network model consisting of a single RTL layer [Canini *et al.*, 2016] from the TensorFlow Lattice library, which is a random ensemble of tiny lattices (i.e., TL Lattices). This layer consists of $\lceil d/r \rceil$ tiny lattices, and each tiny lattice receives an $r$-dimensional input vector, where $r$ is an integer hyperparameter. Hence, this layer uses $O((d/r)k^r)$ space, which is significantly smaller than that of TL Lattice when $r \ll d$. Note that a theoretical analysis in [Canini *et al.*, 2016] shows that TL RTL with $d/r$ tiny lattices is equivalent to the single $d$-dimensional TL Lattice. We used this model for datasets with high-dimensional inputs (i.e., $d$ is a large integer).

We used Python 3.8.12 and PyTorch 1.8.1 to implement MLP and HLL, and the implementation of HLL is available at `https://github.com/IBM/pmlayer`. As for TL Lattice and TL RTL, we used TensorFlow 2.3.0 and TensorFlow Lattice 2.0.10, which is the implementation provided by the authors [Canini *et al.*, 2016; Gupta *et al.*, 2016] and available as an open source library under Apache License 2.0 at `https://www.tensorflow.org/lattice`. The TensorFlow Lattice library also provides an implementation of the Kronecker-Factored lattice layer [Morioka *et al.*, 2021], which uses a smaller amount of memory than TL Lattice. However, we did not include the experimental results for this layer because TL RTL significantly outperformed it for the datasets we used in terms of prediction performance.

Throughout our experiments, all problems were solved as regression problems, and we used mean squared error (MSE) for the evaluation metric. For the training of the neural network models, we used the Adam optimizer [Kingma and Ba, 2015]. In HLL, TL Lattice, and TL RTL, the number of vertices in each dimension of the hypercube was set to $k$ by default. However, the numbers of the vertices in the dimensions that correspond to small categorical values were set to the numbers of the categories. For example, if the $i$-th dimension corresponds to a binary feature, then we set $k_i = 2$ instead of $k_i = k$.

## 4.2 Real Datasets

We used 12 real datasets taken from the UCI Machine Learning Repository [Dua and Graff, 2017] in our experiments. We used six small datasets: Energy Efficiency [Tsanas and Xifara, 2012], QSAR Aquatic Toxicity [Cassotti *et al.*, 2014], Concrete Compressive Strength [Yeh, 1998], Contraceptive Method Choice [Lim *et al.*, 2000], Abalone [Nash *et al.*, 1994], and Shill Bidding [Alzahrani and Sadaoui, 2020], and six large datasets: Online Shoppers Purchasing Intention [Sakar *et al.*, 2019],

Table 1: Datasets obtained from UCI Machine Learning Repository.

| Name | $n$ | $d$ | Monotone features |
|---|---|---|---|
| Energy Efficiency | 768 | 8 | X3, X5, X7 |
| QSAR Aquatic Toxicity | 908 | 6 | MLOGP, SM1_Dz(Z) |
| Concrete Compressive Strength | 1 030 | 8 | *Water* |
| Contraceptive Method Choice | 1 473 | 12 | Number of children ever born, *wife age* |
| Abalone | 4 176 | 10 | Shell, *shucked*, *viscera*, and whole weights |
| Shill Bidding | 6 321 | 9 | Winning_Ratio |
| Online Shoppers Purchasing | 12 330 | 74 | *BounceRates*, *ExitRates* |
| Adult | 30 162 | 88 | Age |
| Online News Popularity | 39 644 | 58 | n_tokens_content, n_tokens_title |
| Facebook Comment Volume | 40 949 | 52 | Column 37 |
| Bank Marketing | 41 188 | 63 | cons.conf.idx |
| Blog Feedback | 46 864 | 276 | Columns 51–53 |

Adult, Online News Popularity [Fernandes *et al.*, 2015], Facebook Comment Volume [Singh *et al.*, 2015], Bank Marketing [Moro *et al.*, 2014], and Blog Feedback [Buza, 2014]. We confirmed that none of these datasets contained any personally identifiable data points nor any offensive content.

Table 1 shows a summary of these 12 datasets. Some of the datasets were already split into training and test data, and the others were unsplit (i.e., a single dataset was given). Column $n$ of this table shows the number of data points in the training dataset for the split datasets and the number of data points for the unsplit datasets. In our experiments, we removed all the data points that had missing feature values.

Column $d$ of Table 1 shows the number of features after our feature encoding. We briefly explain our feature encoding here. We normalized all the continuous values (including the target values) to a range between 0 and 1 by using the min-max scaling algorithm (i.e. a simple linear transformation). We used label encoding to encode the ordinal categorical values. For example, we replaced the ordinal categorical values $\{low, med, high\}$ with $low = 0$, $med = 0.5$, and $high = 1$ so that the replaced values were in the range between 0 and 1. We used one-hot encoding for the other categorical values. When we used TL Lattice and TL RTL, we renormalized the input feature vectors from $[0, 1]^d$ to $[0, k - 1]^d$, where $k$ is a hyperparameter, to conform to the input specifications of these two models.

The last column of Table 1 shows the monotone features we used in our experiments. The *italic* feature names represent monotonically decreasing features, and the others represent monotonically increasing features. We should choose monotone features by using domain knowledge for each dataset, but we used the SHAP values [Lundberg and Lee, 2017] to find candidates of monotone features for the datasets for which we had little domain knowledge. We expected that there would be an almost monotonic relationship between the SHAP value and the $i$-th feature value (i.e., $x[i]$) if the target value was monotonic on the $i$-th feature. In our experiments, we constructed an MLP model for each dataset and computed its SHAP values. Then, we checked for each feature if the SHAP values were monotonically increasing (or decreasing) with respect to the feature values by using Spearman's rank correlation coefficients. Note that such correlation between SHAP values and feature values does not mean a monotone relationship, but we found this approach helpful to find candidates of monotone features for datasets with little domain knowledge.

### 4.3 Experimental Results

In our experiments, we split the data points into training, validation, and test data points. For the unsplit datasets, we divided the data points into training (60%), validation (20%), and test (20%). We used a single random split to search for the best hyperparameters and five random splits to obtain the average prediction performance. For the datasets that were already split into training and test datasets, we further divided the data points in the training dataset into training (80%) and validation (20%) and kept the test dataset unchanged. We used a single random split to search for the best hyperparameters and five random splits to obtain the average prediction performance without changing the test dataset.

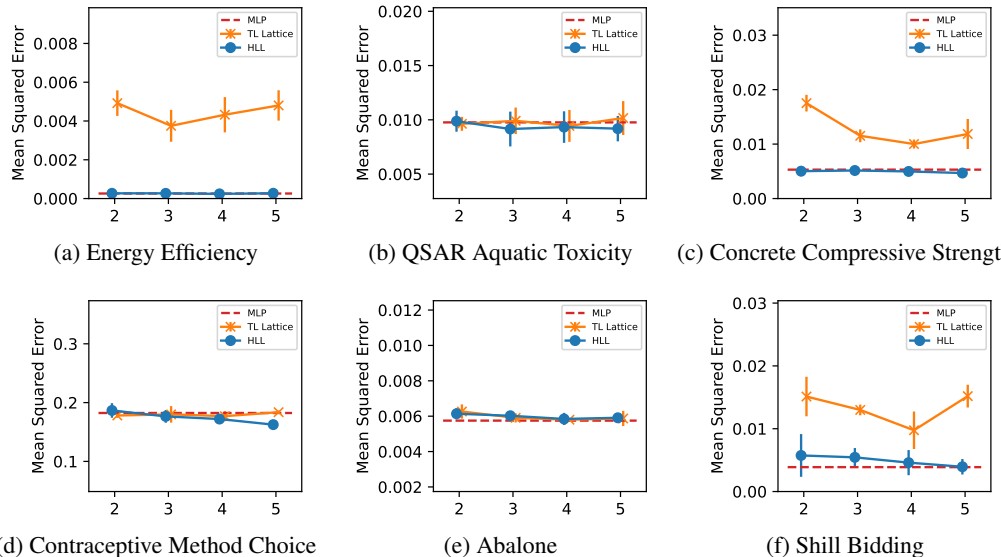

Figure 4: MSE (lower is better) of the neural network models with various $k$ on small datasets.

Regarding the hyperparameters, we chose the best hyperparameters for each combination of a neural network and a dataset; the learning rate was chosen from $\{1, 10^{-1}, 10^{-2}, 10^{-3}, 10^{-4}, 10^{-5}\}$, the batch size was chosen from $\{8, 16, 32, ..., 4096\}$, the number of neurons in the hidden layers was chosen from $\{16, 32, 64, ..., 512\}$ for MLP and HLL, and the hyperparameter $r$ was chosen from $\{4, 5, 6, 7\}$ for TL RTL. We used only the training and validation datasets to determine the best hyperparameters, and we used Optuna [Akiba *et al.*, 2019] to search for the best hyperparameters.

**Results for Small Datasets.** We compared the prediction performances of MLP, HLL, and TL Lattice by using the six small datasets (i.e., $d$ was small). We trained these neural network models for 1000 epochs with certain early stopping criteria, and Figure 4 shows the results for various $k$. The red dashed lines show the average of the MSEs for five runs of MLP, and we regard these values as baselines of the prediction performance. The blue and orange lines show the average MSEs together with their standard deviations for five runs of HLL and TL Lattice, respectively. These results showed that our HLL performed comparably to MLP for all six datasets. Somewhat surprisingly, we note that using a finer-grained lattice (i.e., increasing $k$) did not necessarily improve the MSEs. This phenomenon was already mentioned in [Gupta *et al.*, 2016] for TL Lattice, and we observed this phenomenon both for TL Lattice and HLL in our experiments. Figure 4 also shows that the prediction performance of HLL were also comparable with TL Lattice. Although TL Lattice showed worse MSEs than MLP and HLL for some datasets, these differences in prediction performance were supposed to come from two factors. One was the difference of the parameterization; HLL used a hypercube of size $k^m$, whereas TL Lattice used a hypercube of size $k^d$. The other one was the difference of the neural network training algorithm; MLP and HLL used the standard stochastic gradient descent algorithm, whereas TL Lattice used the projected gradient descent algorithm. In addition, we compared the prediction performace of HLL and TL Lattice with the calibration layer, and the results are shown in Appendix A.4.

**Results for Large Datasets.** For the datasets with a large $d$, we could not use TL Lattice due to its space complexity $O(k^d)$, so we used TL RTL instead. TL Lattice requires that at least $2^{52}$ parameters be stored even if $k = 2$ since $d \geq 52$, but we cannot store such a large number of parameters in memory. In contrast, we could use HLL since $m$ is small in the datasets we used.

We compared the prediction performance of MLP, HLL, and TL RTL by using the six large datasets. We trained these neural network models for 100 epochs with certain early stopping criteria, and Figure 5 shows the results for various $k$. The red dashed lines show the average MSEs for five runs of MLP, and we regard these values as baselines of the prediction performance. The blue and green lines show the average MSEs together with their standard deviations for five runs of HLL and TL

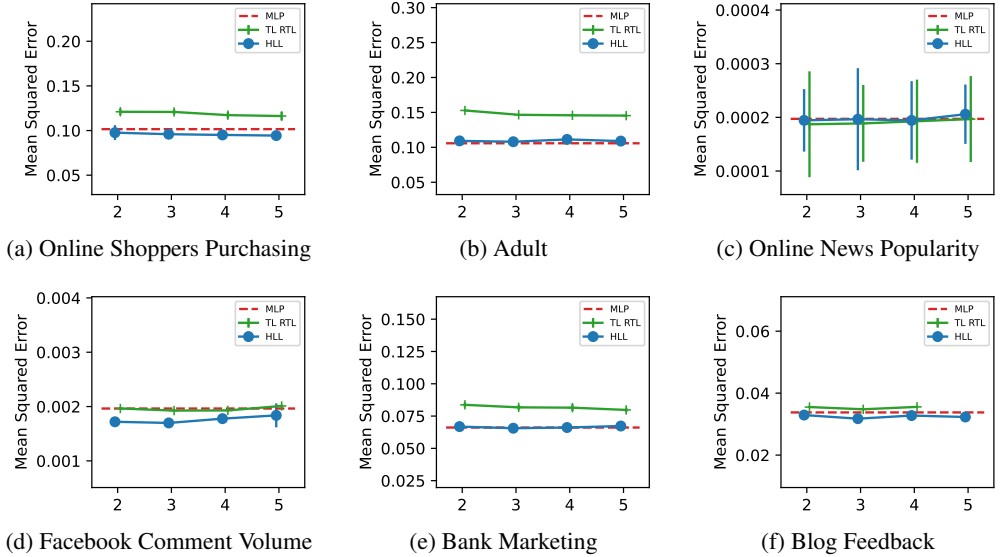

Figure 5: MSE (lower is better) of the neural network models with various $k$ on large datasets.

RTL, respectively. Note that the result for TL RTL with $k = 5$ on the Blog Feedback dataset was missing due to a memory limitation in our computer environment. These results showed that our HLL performed comparably to MLP and TL RTL regardless of $k$ for the six large datasets. We think that HLL performed better than MLP for some datasets thanks to the regularization effects coming from the monotonicity constraints.

## 5  Related Work

There are some neural networks for partially monotone regression other than the TensorFlow Lattice library. One of the simplest approaches is to enforce non-negativity constraints on neural network weights that are related to monotone inputs (e.g., [Daniels and Velikova, 2010; You *et al.*, 2017]). Another simple approach is to use a regularization term to enforce monotonicity [Gupta *et al.*, 2019; Monteiro *et al.*, 2021], but a prediction model that uses this approach may not always satisfy the monotonicity constraints. More complex approaches are proposed in [Liu *et al.*, 2020; Sivaraman *et al.*, 2020]. These approaches use a mixed integer linear programming solver or a satisfiability modulo theories (SMT) solver to ensure the monotonicity of the neural network. The disadvantages of these approaches are their computational complexity and the costs of using the solvers.

## 6  Conclusion

We proposed the hierarchical lattice layer (HLL) for partially monotone neural networks as an extension of TL Lattice [Gupta *et al.*, 2016]. We demonstrated that HLL resolved two problems with TL Lattice: the memory consumption and the requirement for the projected gradient descent algorithm with many constraints. In our experiments, we showed that HLL did not sacrifice prediction performances to resolve these problems. Note that our experiments were limited to a comparison between single layers (i.e., HLL and TL Lattice). The prediction performance of these models might be able to be improved by combining multiple layers (e.g., [You *et al.*, 2017]) or any other techniques (e.g., changing feature encoding and increasing the number of layers in MLP) and the memory consumption might also be reduced by using smaller lattices, but note that the requirement for the projected gradient descent algorithm with many constraints remains for TL Lattice.

A limitation of our HLL is that it cannot be used for the applications with a large $m$. If $m$ is large, we need to combine small HLL layers to construct a large neural network as demonstrated in [Canini *et al.*, 2016; You *et al.*, 2017] for TL Lattice.

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
