# A  Experiments

## A.1  1D Synthetic Datasets

We show our experimental results on one-dimensional synthetic datasets $D_{\text{square1}} = \{(x_i, y_i)\}_{i=1}^n$ such that $y_i = x_i^2$ and $D_{\text{sqrt1}} = \{(x_i, y_i)\}_{i=1}^n$ such that $y_i = \sqrt{x_i}$. Figures 1 and 2 show the prediction results with various $k$ for these two datasets. The blue lines show the predictions by HLL, and the black lines show the ground truth. Both figures show that increasing $k$ yields better prediction results on these synthetic datasets.

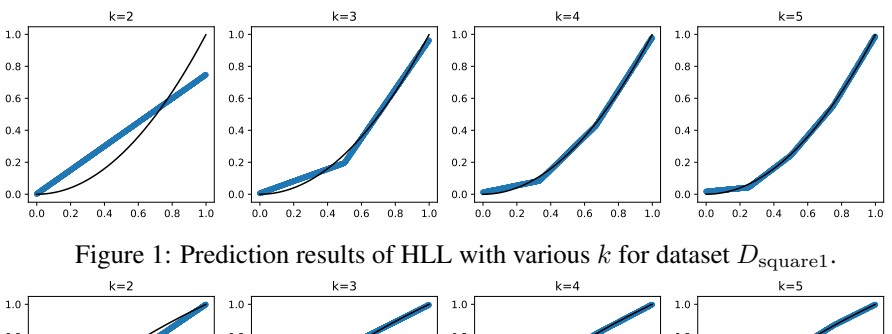

Figure 1: Prediction results of HLL with various $k$ for dataset $D_{\text{square1}}$.

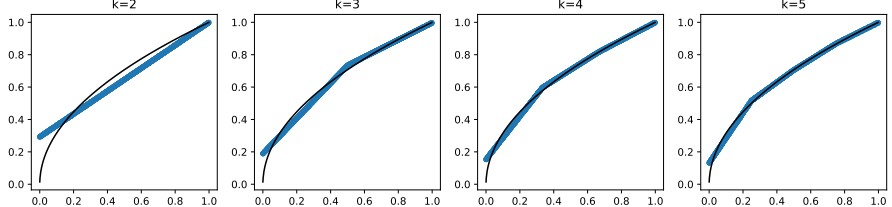

Figure 2: Prediction results of HLL with various $k$ for dataset $D_{\text{sqrt1}}$.

## A.2  2D Synthetic Datasets.

We show our experimental results on two-dimensional synthetic datasets $D_{\text{square2}} = \{(x_i, y_i)\}_{i=1}^n$ such that $y_i = (x_i[1]^2 + x_i[2]^2)/2$ and $D_{\text{sqrt2}} = \{(x_i, y_i)\}_{i=1}^n$ such that $y_i = (\sqrt{x_i[1]} + \sqrt{x_i[2]})/2$ (recall that the $i$-th element of vector $x$ is denoted as $x[i]$). Figures 3 and 4 show the prediction results of HLL, TL Lattice, and TL RTL with $k = 4$ against the ground truth. These prediction models show similar prediction results on these synthetic datasets.

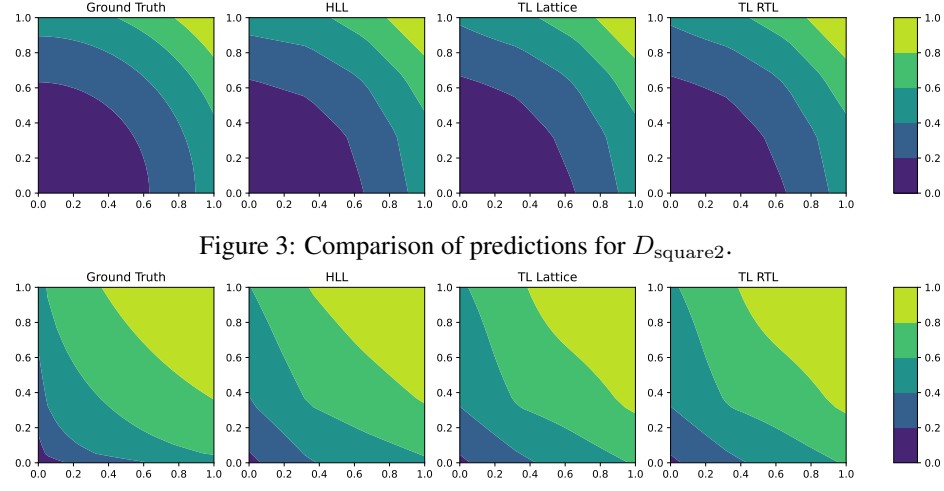

Figure 3: Comparison of predictions for $D_{\text{square2}}$.

Figure 4: Comparison of predictions for $D_{\text{sqrt2}}$.

## A.3 Training Time

We compared the training times of HLL and TL Lattice on the small datasets. In these experiments, we fixed the hyperparameters; $k$ was set to 2, the learning rate was set to $0.001$, and the batch size was set to $128$. The number of epochs was set to 100, and we did not use any early stopping. Figure 5 shows the results, where we varied the number of neurons in the auxiliary neural network $g_\theta(p, v)$ from 16 to 512 for HLL, and the execution times were the averages of five runs. These results show that the training times of HLL were shorter than those of TL Lattice for these six small datasets.

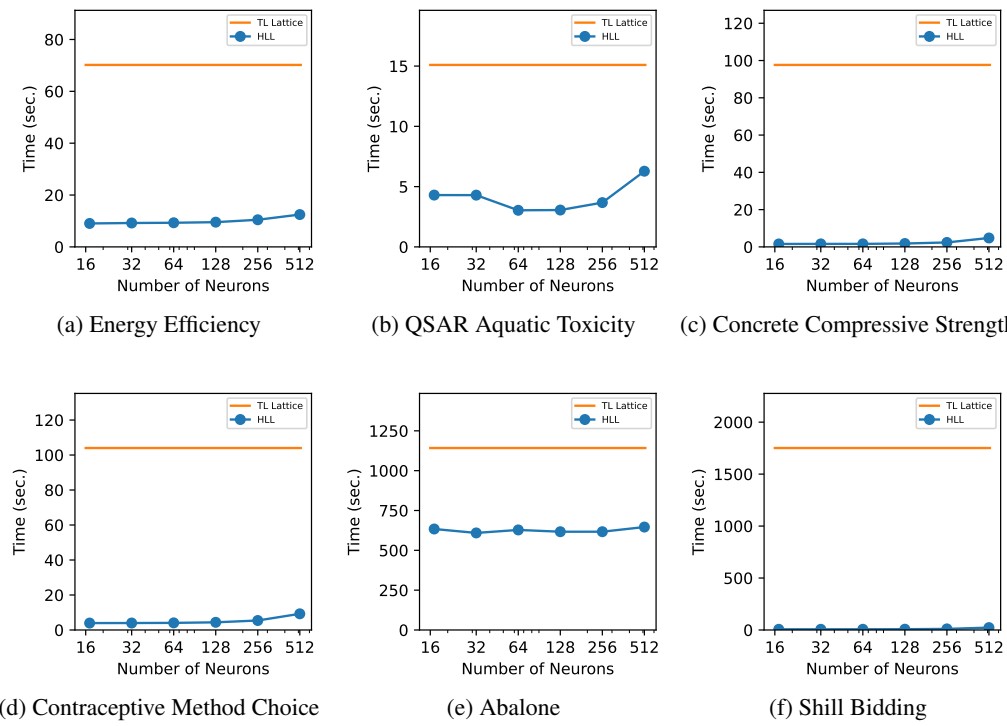

Figure 5: Comparison of training times (in sec.) of neural network models for various numbers of neurons of HLL on small datasets.

## A.4 Combination with Calibration Layers

We conducted additional experiments by using the calibration layers [You *et al.*, 2017]. More specifically, we added the calibration layer for each dimension of the input $x$ before the input of the lattice layer and the monotonicity constraint were imposed on the calibration layers for the monotone inputs. The number of the end points in each calibration layer was set to 100 and the end points were uniformly distributed over the input domain. We used both the multilinear interpolation and the simplex interpolation for TL Lattice.

Figure 6 shows the results, where the red dashed lines show the average of the MSEs for five runs of MLP. The blue, orange, and brown lines show the average MSEs together with their standard deviations for five runs of HLL, TL Lattice with multilinear interpolation, and TL Lattice with simplex interpolation, respectively. These results showed that the prediction performance of TL Lattice with the calibration layers were improved for the three datasets (a) Energy Efficiency, (c) Concrete Compressive Strength, and (f) Shill Bidding compared with the results shown in Figure 4. However, they were still not better than HL Lattice. We also note that there were no clear advantage of the simplex interpolation compared to the multilinear interpolation with respect to MSEs.

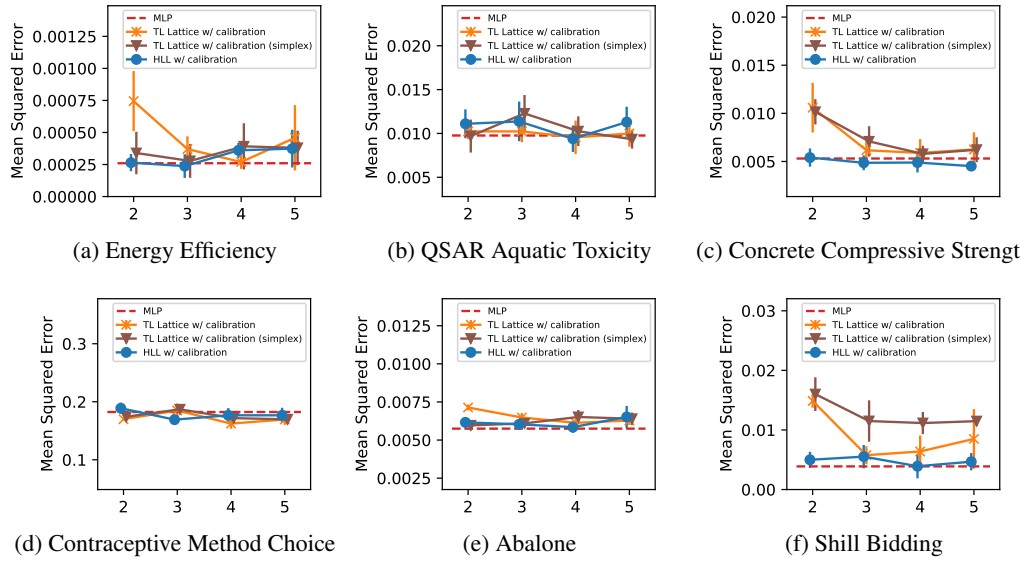

Figure 6: MSE (lower is better) of the neural network models with various $k$ on small datasets.

## B  Multilinear Interpolation

For the completeness of this paper, we briefly describe multilinear interpolation algorithm [Gupta *et al.*, 2016] here. For simplicity, we assume that we are given a vector $q \in [0, 1]^m$, and the set of vertices that surround $q$ is the set of vertices of the (undivided) unit hypercube $V_q = \{v \mid v \in \{0, 1\}^m\}$. Assuming that we know the value $f(p, v)$ for any $v \in V_q$ and $p \in \mathbb{R}^{d-m}$, we can compute the multilinear interpolation by

$$f(p, q) = \sum_{v \in V_q} f(p, v) \prod_{i=1}^{m} q[i]^{v[i]} (1 - q[i])^{1-v[i]},$$

where $q[i]$ and $v[i]$ represent the $i$-th dimension of vectors $q$ and $v$, respectively.

**Example.**   We can compute $f(p, q)$ for $q = (0.3, 0.6)$ by

$$\begin{aligned} f(p, q) &= f(p, v_{(0,0)}) \cdot (1 - 0.3)(1 - 0.6) + f(p, v_{(0,1)}) \cdot ((1 - 0.3) \cdot 0.6) \\ &\quad + f(p, v_{(1,0)}) \cdot (0.3 \cdot (1 - 0.6)) + f(p, v_{(1,1)}) \cdot (0.3 \cdot 0.6) \end{aligned}$$

when $m = 2$ and $V_q = \{v_{(0,0)}, v_{(0,1)}, v_{(1,0)}, v_{(1,1)}\}$.