# OpenReview forum: "Hierarchical Lattice Layer for Partially Monotone Neural Networks"
_NeurIPS.cc/2022/Conference — NeurIPS 2022 Accept_

### Official Review · Reviewer_hkrw · 2022-07-06

**Rating:** 7
**Confidence:** 3
**Soundness:** 4 excellent
**Presentation:** 2 fair
**Contribution:** 3 good

**Summary:**

This paper proposes a novel neural net layer that improves on the TensorFlow lattice layer by being able to take advantage of standard neural net training procedures (the existing lattice layer requires a special training procedure) and also being able to handle high-dimensional inputs (whereas the existing lattice layer would run into memory issues).

**Questions:**

Please address the points that I listed as weaknesses above.

**Limitations:**

Yes, the author(s) do briefly address the limitation of their approach (i.e., it doesn't handle large $m$), and I found their response to the question on potential negative societal impact in the checklist to be adequate.

**Strengths And Weaknesses:**

I would say that overall, the technical advance is very incremental. For example, much of the same ideas are already used in survival analysis for estimating the monotonically decreasing survival probability function over time (e.g., Kvamme and Borgan 2021, although the same ideas in parameterizing the monotonic survival function appear in earlier papers, some of which are referenced within Kvamme and Borgan), where standard neural net training procedures suffice. One could perhaps view the proposed HLL method as a generalization of what is already done in survival analysis that does not need a specialized training procedure (although really I'd be surprised if the same ideas used by survival analysis researchers haven't already been used by researchers in other fields that need to enforce monotonicity). That said, I think a paper that neatly shows that there's a general idea that subsumes existing approaches is valuable.

Strengths:
- The key technical ideas are actually not that complicated although the presentation perhaps make them sound more complicated than they actually are.
- The author(s) provide sufficient empirical evidence that the proposed hierarchical lattice layer (HLL) is better than the existing TensorFlow lattice layer in the case of partially monotone regression.

Weaknesses:
- I found the presentation at times to be more complicated than it needs to be. I would suggest adding a simple running example (could be very low-dimensional) throughout the paper that already clearly shows why the proposed method clearly works and we really don't need a specialized training procedure.
- It would be helpful relating the proposed method to possibly special cases of how people have previously enforced monotonicity without resorting to specialized training procedures (I mentioned already that this is done in survival analysis but I'd assume that it's also done in other fields too).
- Minor: Please appropriately switch between using "citep" and "citet" for references (currently, the paper basically uses the equivalent of "citet" too often so that the text suddenly switches to author names in a manner that is not grammatically correct and detracts from reading the paper). As a few examples:
    - In line 19-20, I'd suggest you use "citep" so that the text says "... if the other features are equal [Potharst and Feelders, 2002]."
    - In lines 25-28, if you use "citep", you would get: "For example, they have a better regularization capability [Dugas et al., 2000; Fard et al., 2016; You et al., 2017] and better interpretability [Gupta et al., 2016], and they can be used for fair machine learning [Wang and Gupta, 2020]."

Reference:

Kvamme H, Borgan Ø. Continuous and discrete-time survival prediction with neural networks. Lifetime Data Analysis. 2021 Oct;27(4):710-36.

---

> ### Author Response · Authors · 2022-08-02
> **Answers to your questions.**
>
> First of all, thank you for the paper [Kvamme and Borgan 2021].  This paper describes the standard approach for survival analysis, and the algorithm in Section 3.3 seems somewhat similar to the calibration layer in Tensorflow Lattice [You et al., 2017].  However, we are afraid to say that we could not find the same ideas with ours.  We would appreciate if you could elaborate on the relationship between these two papers.
>
> > I found the presentation at times to be more complicated than it needs to be. I would suggest adding a simple running example (could be very low-dimensional) throughout the paper that already clearly shows why the proposed method clearly works and we really don't need a specialized training procedure.
>
> Thank you for your suggestion.  We admit that our presentation is a bit complicated.  One of the causes of the complexity comes from the explanation on the space required to store $L_{v,\sigma}$ and $U_{v,\sigma}$ is $O(m|V_{m}|)$ (in Lines 196-201), although we know that the presentation of our paper can be simplified if we can use $O(|V_{m}|^2)$ space.  We will consider more easy-to-understand explanation of our model by utilizing Appendix A.
>
> > Minor: Please appropriately switch between using "citep" and "citet" for references (currently, the paper basically uses the equivalent of "citet" too often so that the text suddenly switches to author names in a manner that is not grammatically correct and detracts from reading the paper).
>
> Thank you for spotting this.  We will fix this problem.

---

> > ### Comment · Reviewer_hkrw · 2022-08-05
> > **response to author response**
> >
> > Thanks for the response.
> >
> > Survival analysis setup: I was hoping that you'd be able to elaborate on the relationship since I get the impression that the survival analysis case really is a super simple special case (but maybe I'm not understanding things correctly?). I'm not sure if there are other survival analysis papers that make the setup clearer. Let me try to explain. Basically in survival analysis, the survival function to be estimated monotonically decreases as it is 1 minus a CDF. This function is commonly parameterized in terms of discrete time steps (could be all unique observed times in the training set or a user-specified grid, etc). The grid is of course extremely simple as it's just on a 1D line (so this is a really, really simple lattice). Then commonly people either convert the survival function to a discrete probability mass function (PMF) along the grid (or they convert it to be the discrete hazard function; the Kvamme and Borgan paper describes straightforward ways to convert between the survival function/PMF/discrete hazard function representations). For instance, given a PMF, if you do a cumulative sum in reverse-chronological order, then you recover the survival function. The PMF is then parameterized so that each entry in it along the lattice is the output of, for instance, a sigmoid function or some other activation (that guarantees nonnegative outputs) that neural net frameworks trivially handle. As it is straightforward to convert between the PMF/discrete hazard/survival function representations, it is easy to use different standard differentiable survival analysis losses in a neural net framework (standard nonnegative log likelihood can be used, for instance). No special training procedure is needed (note that because this is so straightforward in survival analysis, to the best of my knowledge, survival analysis papers nowadays do not actually say that no special training procedure is required as it is obvious). From my understanding, your paper trivially handles this setup as well as substantially more complicated settings with higher-dimensional lattices. The idea for why no special training procedure is needed seems to basically be the same?
> >
> > Clarity of presentation: I realize that it can be take some work getting the exposition to be as simple and clear as possible while getting the most efficient time/space complexity. I'm usually in favor of going over the easier-to-understand version first and saying that there's a refinement to reduce the complexity, but I realize that this is stylistic.

---

> > > ### Author Response · Authors · 2022-08-07
> > > **Thank you for your explanation.**
> > >
> > > Thank you for your explanation on the paper [Kvamme and Borgan 2021].
> > >
> > > We think that it is very usual to use a softmax function to estimate a discretized probability distribution (as shown in Equation (9) in [Kvamme and Borgan 2021]) and therefore we did not think that we should emphasize that no special training algorithm is required for such a simple task.
> > >
> > > However, we now noticed that this paper also presents another approach by estimating the hazard rate $h(\tau_{j} | x)$ by using a sigmoid function in Section 2.2 and that this approach can be seen as a special case of ours for the case $m=1$.
> > >
> > > We will discuss this issue when we update our paper (if this paper is accepted).  Thank you for your clarification.

---

> > > > ### Comment · Reviewer_hkrw · 2022-08-09
> > > > **updating my score**
> > > >
> > > > Great, glad we cleared this up! In consideration of all reviews and the discussion thus far, I will update my score to "7: Accept".

---

### Official Review · Reviewer_utRT · 2022-07-11

**Rating:** 7
**Confidence:** 5
**Soundness:** 4 excellent
**Presentation:** 3 good
**Contribution:** 3 good

**Summary:**

This paper introduces a new model architecture to represent partially monotonic functions that better scale to a larger number of inputs compared to current baselines. Monotonic functions are ubiquitous in real-world ML applications, with many examples included in the paper. The new technique is based on a reparametrization of lattice models (interpolated lookup tables). Instead of explicit parametrization of Gupta et al. [2016], the proposed method uses an auxiliary (non-monotonic) neutral network to linearly interpolate in the space of upper and lower bounds of each lattice site/vertex. The auxiliary model is thus implicitly defining the value of the vertices on the lattice while satisfying monotonicity constraints.

Note: I reviewed an earlier version of this work. The new submission is improved in several aspects, including the presentation of the method and empirical studies.


**Questions:**

Questions:
* The specific parametrization of the lattice described in the work results in a compound influence of the first output of the auxiliary model through the hierarchy of calculations, whereas the last output of the auxiliary model is only used in the calculation of the value of the last vertex. One would expect the gradients to flow very differently in the backward path for different outputs of the auxiliary model (and possibly issues vanishing gradient if D is large). Did the authors notice any significant effects related to this asymmetry?

Notes and suggestions:
* The citations in the paper have incorrect formatting. The authors should choose the correct citation latex macros that would include parentheses when inlined in text.
* IMO, “Explicitly Parameterized Lattice” (EP Lattice) might be a better name for the baseline.
* Projected gradient descent used for baseline lattice models is very much a standard training algorithm, with projections supported as first class citizens in several ML libraries (e.g. TF/Keras, PyTorch). The calculation of the projection is of course non-trivial and specific to the modeling. It would be better if the authors are more explicit about this distinction in their description of the baseline training algorithms.
* Line 170: … hold for any ordering sigma: … any ordering sigma consistent with the partial ordering defined by the monotonicity constraints.
* Please check plots in a black and white print. Some line formattings are hard to tell apart at the current resolution.


**Limitations:**

IMO the authors have sufficiently discussed the limitations of their work.

**Strengths And Weaknesses:**

Strength:
* The lattice reparametrization idea presented in the paper is very interesting, as it makes it possible to model monotonic functions via a generic non-monotonic auxiliary function. This makes it possible to use generic off-the-shelf models on non-monotonic features (e.g. a CNN applied to an input image), while making sure the model is monotonic w.r.t. other features (e.g. quantile value in a quantile regression task).
* The proposed implicit representation of the lattice parameters forces monotonicity by construction. This avoids the need for the expensive parameter projection step during training, which is needed for explicitly parameterized lattice models.
* The method should be easy to implement in any standard ML framework.


Weaknesses:
* The calculation of vertex values in the proposed method adds extra compute compared to explicit representations when the number of features in each lattice is small. If simplex interpolation is used (Gupta et al. [2016]), the extra computation needed for the proposed method will change the overall evaluation time from O(D logD) to O(K^D) for D features used in a lattice with K vertices along each dimension. This is not a big concern since the main use case argued in the paper is when only a small number of features are monotonic, and the proposed method only needs to apply lattice interpolation to the monotonic features. However, I believe simplex interpolation could have been studied in the empirical studies at least for the baseline lattice models.
* Unlike the studies of baseline lattice models (Gupta et al. [2016], Fard et al. [2016], You et al. [2017], Wang and Gupta [2020]), empirical studies in this paper do not use a trainable calibration layer before passing the input to the lattice. Since the vertices of the lattices are fixed at equidistance locations, standard linear normalization is likely suboptimal for features with non-uniform distributions. Increasing the lattice resolution (as the paper concurs with previous work) will not solve the issue since it is adding parameters uniformly across the input domain where they might not be needed.

---

> ### Author Response · Authors · 2022-08-02
> **Answers to your questions**
>
> > Unlike the studies of baseline lattice models (Gupta et al. [2016], Fard et al. [2016], You et al. [2017], Wang and Gupta [2020]), empirical studies in this paper do not use a trainable calibration layer before passing the input to the lattice.
>
> > However, I believe simplex interpolation could have been studied in the empirical studies at least for the baseline lattice models.
>
> As you suggested, we conducted the experiments using the calibration layer and the simplex method in Tensorflow Lattice for the small datasets, and the results are included as Appendix E in the updated manuscript (note that the other sections are unchanged).  As you expected, the MSEs of the TL Lattice were reduced by combining the calibration layer and TL Lattice, but our HLL model was still competitive or better than the TL Lattice with calibration layer.
>
> We further changed the interpolation method of TL Lattice to the simplex method, but we could not see significant improvements over the multilinear interpolation.  Note that the original paper [Gupta et al., 2016] also tells that the prediction performances of the multilinear and simplex interpolation are similar.
>
>
> > Did the authors notice any significant effects related to this asymmetry?
>
> Not really.  We did not notice any problems in the training of our neural network models.
>
> > The citations in the paper have incorrect formatting.
>
> Thank you for spotting this.  We will fix this problem.
>
> > Projected gradient descent used for baseline lattice models is very much a standard training algorithm, with projections supported as first class citizens in several ML libraries (e.g. TF/Keras, PyTorch).
>
> Thank you for your comments.  We will update the related sentences taking into account your comments.
>
> > Line 170: … hold for any ordering sigma
>
> This phrase is correct as it is, because the monotonicity constraints are satisfied as long as $L_{\sigma,v}$ and $U_{\sigma,v}$ are appropriately calculated and we can use any ordering $\sigma$ here.  For example, we can use $\sigma=(v_{2}, v_{0}, v_{1})$ or $\sigma=(v_{1}, v_{0}, v_{2})$ instead of $\sigma=(v_{0}, v_{1}, v_{2})$ for the case in Appendix A.
>
> > Please check plots in a black and white print.
>
> Thank you for spotting this.  We will update the line charts taking into account the black and white print.

---

> > ### Comment · Reviewer_utRT · 2022-08-08
> > **Reviewer Response**
> >
> > I thank the authors for including the requested experiment with added calibrators. Since the results are clearly improved for the baseline model, please refer to them in the main body of the paper.
> >
> > The authors have sufficiently addressed my questions and concerns. I am keeping my accept recommendation.

---

> > > ### Comment · Reviewer_utRT · 2022-08-09
> > > **Appendix E**
> > >
> > > For better reproducibility, please add for the following details to the experiments with calibrators:
> > > - if the calibrators were constrained to be monotonic (they should be to achieve end-to-end monotonicity)
> > > - were the keypoints of the calibrators spread uniformly in the input space, or did you use feature quantiles (as in [You et al., 2017])?

---

### Official Review · Reviewer_GeFP · 2022-07-16

**Rating:** 7
**Confidence:** 4
**Soundness:** 3 good
**Presentation:** 3 good
**Contribution:** 2 fair

**Summary:**

This work proposes a novel neural network layer to model partial monotonicity in regression analysis. To enforce monotonicity w.r.t. a subset of input features, the authors define a hierarchical lattice structure over a unit hypercube spanning the space of monotonic features. The authors then employ an ordered set (based on the coordinates) of vertices of the lattice structure to define upper and lower bounds of the target regression function for a given set of input features. Given the upper and lower bounds corresponding to an input, the authors propose to approximate the target function by using interpolation between the bounds. They learn an auxiliary neural network with sigmoid head to output the interpolation weight given input features. The authors perform complexity analysis of the proposed method and in the numerical experiments, they demonstrate that their method is competitive or better than compared methodologies, which also includes a generic multi-layer perceptron model.

**Questions:**

Have you analysed if for instance in your experiments a generic model like an MLP also learns (partially) monotonic relationships between input and target variables, even though it is not enforced by design?

**Strengths And Weaknesses:**

Strengths:
- Compared to previous approaches, the proposed method reduces the spatial complexity to the dimensionality of monotonic features
- The lattice layer can be trained using standard optimisation methods such as back propagation.
- Numerical results are competitive or better than compared approaches

Weaknesses:
- The approach requires prior knowledge of monotonic features and the layer cannot yield optimal results if monotonicity assumptions do not hold.
- Time and space complexity of the proposed method increases exponentially w.r.t. the lattice granularity and the number of monotonic features, hence it's application to a large number of monotonic features can become infeasible and the performance may suffer due to a coarse granularity of the lattice structure.
- Although the method is competitive to generic methods like MLP, it can be simply down to the choice of architecture used in the comparison.

---

> ### Author Response · Authors · 2022-08-02
> **Answer to your question.**
>
> > Have you analysed if for instance in your experiments a generic model like an MLP also learns (partially) monotonic relationships between input and target variables, even though it is not enforced by design?
>
> We have not verified it in our experiments.  However, in quantile regression (i.e., partially monotone regression with $m=1$), it is widely known that the monotonicity constraints are sometimes not satisfied if we use the standard MLP, which is known as the crossing problem.  (Note that , in quantile regression, the output must be monotonically increasing with respect to quantile level $0 \leq \tau \leq 1$ for input $x=(x',\tau)$.)  An experimental example of the crossing problem in quantile regression can be found in Figure 2 of [Tagasovska and Lopez-Paz, 2019].
>
> [Tagasovska and Lopez-Paz, 2019]: Natasa Tagasovska and David Lopez-Paz, "Single-Model Uncertainties for Deep Learning," NeurIPS 2019.

---

> > ### Comment · Reviewer_GeFP · 2022-08-09
> > **Response to authors**
> >
> > I thank authors for answering my question. Taking into account other reviews and authors responses, I am updating my score to accept.
> >
> > It would suggest that authors add reference to the crossing problem example in [Tagasovska and Lopez-Paz, 2019] to the updated version of their manuscript.

---

### Meta-Review · Area_Chair_LuQv · 2022-08-26

**Recommendation:** Accept
**Confidence:** Certain

**Metareview:**

Overall, the reviews about this paper are very positive. The authors spent great effort engaging in discussions and improving the paper with clarifications and additional experiments. We recommend accepting the paper.

**Award:**

No

---

### Decision · Program_Chairs · 2022-09-14

Accept